# Effect of Shungite Application on the Temperature Sensitivity of *Allium cepa* Respiration under Two Soil Water Regimes †

Elena Ikkonen [1,*], Svetlana Chazhengina [2] , Olga Bakhmet [3] and Valeria Sidorova [1]

1. Karelian Research Center RAS, Institute of Biology, Puskinskaja, 11, 185610 Petrozavodsk, Russia; val.sidorova@gmail.com
2. Karelian Research Center RAS, Institute of Geology, Puskinskaja, 11, 185610 Petrozavodsk, Russia; chazhengina@mail.ru
3. Karelian Research Center RAS, Puskinskaja, 11, 185610 Petrozavodsk, Russia; bahmet@krc.karelia
* Correspondence: likkonen@gmail.com; Tel.: +7-911-0543-124
† Presented at the1st International Electronic Conference on Agronomy, 3–17 May 2021; Available online: https://sciforum.net/conference/IECAG2021.

**Abstract:** For agricultural soils with low natural fertility, the possibility of using rock powders as an alternative source of nutrients and/or improver of soil physical parameters is under discussion and study. Shungite rocks, carbon-bearing volcanic sedimentary rock, are characterized by a high content of carbon and nutrients. This study aimed to evaluate whether shungite application to Umbric Podzols may affect leaf and root mitochondrial respiratory pathways, and the leaf response to temperature change. A pot culture experiment was conducted with *Allium cepa* L. seedlings, using soil shungite concentrations of 0, 5, 10, and 20 g kg$^{-1}$ and two soil water regimes: well-watered (WW) and drying-wetting (DW) cycles. Soil water deficit increased total respiration ($V_t$) of onion leaves, but not roots, under low (13 °C) and high (33 °C) measurement temperature. Shungite application affected leaf $V_t$ only at 13 °C: it increased the $V_t$ rate under WW and decreased one under DW. An increase in the measurement temperature to 33 °C enhanced the sensitivity of leaf respiration to the inhibitor of the alternative respiratory pathway (salicylhydroxamic acid, SHAM). Shungite application increased the contribution of SHAM-sensitive pathway to the leaf $V_t$ rate under WW, but not under the DW regime, regardless of the leaf temperature. In contrast to the SHAM-resistant pathway, the temperature sensitivity of the SHAM-sensitive rate decreased following the decrease in soil water availability. Shungite application increased the temperature sensitivity of both SHAM-sensitive and SHAM-resistant pathways under DW, and significantly decreased these parameters under WW. In summary, the decrease in temperature sensitivity of alternative SHAM-sensitive respiratory pathway with a decrease of soil water availability or shungite-related decrease of both SHAM-sensitive and SHAM-resistant leaf respiration may play an important role in enhancing the resistance of plant respiration to stress temperature.

**Keywords:** onion; rock powder; respiratory pathways; soil water deficit

## 1. Introduction

Rocks containing a multitude of nutrients have been proposed as a slow-release fertilizer that allows nutrients to remain in the top soil for a long time [1]. Therefore, the possibility of using rock powders as an alternative source of nutrients for agriculture practice has been widely discussed [2]. When initial nutrient levels in agricultural soils are low, the application of rock powders may improve not only ion and cation exchange capacity [3] but also the physical properties of soils, as was shown for inorganic carbon [4].

Shungite rocks, formed mainly on a silicate basis, are carbon-bearing sedimentary-volcanic rocks widely distributed in the Lake Onega area. The carbonaceous matter characterized by a globular fullerene-like molecular structure is one of the main components of the shungite [5]. The amount of elemental carbon preserved in shungite rocks was

estimated to be more than $25 \times 10^{10}$ tons [5]. Along with inorganic carbon, some macro- and micronutrients, such as Si, K, Ca, Mg, Na, Cu, and others, were found in the shungite rocks [5]. Since most of the nutrient elements are prevalent, soil elements beneficially affect the physiological state of plants, maintaining that adequate plant nutritional status may improve the physiological resistance of plants under stress situations, including stress temperatures [6] and soil water deficit [7].

It is well documented that in agricultural practice not only the low natural fertility of soils but also climatic factors such as stress temperature or an inadequate soil water regime have long been recognized as the main determining stress factors challenging current agricultural productivity. Along with photosynthesis, respiration is one of the main physiological processes responsible for plant growth and development. While a $CO_2$ assimilation rate has been shown to be strongly suppressed by drought [8], the impact of soil water deficit on plant respiration may be multidirectional: it decreases in the initial phase of water stress, and increases, as an acclimation mechanism, under lower soil water availability [9].

Plant respiratory metabolism is altered such that under stress conditions other pathways, besides the cytochrome *c* oxidase (Cyt) pathway, are induced to provide alternative respiratory substrates to the respiratory processes [10,11]. Compelling evidence has recently demonstrated that alternative pathway respiration (Alt) associated with alternative oxidase activity provides flexibility in cellular energy and carbon metabolism, thus contributing to increased resistance of plants to stress conditions including soil water deficit, as well as a low or high temperature [10]. The drought-related increase in plant respiration rate described in [9] can be connected with an enhanced capacity of Alt, as was shown by Feng et al. [12] However, the adjustment of respiratory metabolism, as plant responses to changes of growth conditions, can be connected with increased electron partitioning to the Alt and decreased to the Cyt with the total leaf respiration not affected [13]. A recent study of the effect of both the soil water regime and shungite application on physiological traits of onion seedlings showed that the first one has a much stronger effect on leaf respiration than the second [14]. Although the leaf respiration rate was not affected significantly by shungite application to the soil either under sufficient or low soil water availability, shungite improved the nutrient status of onion leaves and plant resistance to water deficit [14]. This may be due to an impact of shungite on the activity of respiratory pathways, and/or changes in the partitioning between them; however, there has been no published evidence supporting this state. In this study, we hypothesized that earlier reported positive effects of shungite rocks on plant resistance to soil water deficit could be connected with its impact on respiratory pathways.

Plant respiration is a temperature-sensitive process with the temperature sensitivity being referred to as the temperature coefficient ($Q_{10}$), defined as a proportional change in respiration rate per 10 °C change in temperature. The Alt and Cyt pathways have been shown to differ in their sensitivities to short-term changes in temperature [15]. It was proposed that the Alt pathway may maintain mitochondrial electron transport and protect against harmful reactive $O_2$ generation in the cold due to this pathway being less temperature sensitive (lower $Q_{10}$) than the Cyt pathway [16]. However, some studies found few differences in the $Q_{10}$ values between the Alt and Cyt pathways [17] or more sensitivities in the Alt than the Cyt pathway [15,18]. The shift of the temperature sensitivity of these respiratory pathways under changed conditions, for example, soil water or nutrient availability, can alter the partitioning between the pathways, and, consequently, plant resistance to temperature stress.

To estimate the pathway's activity at different temperatures, the respiratory inhibitors are widely used in the studies which investigated the temperature sensitivity of the Alt and Cyt pathways [17,18]. Specifically, the Alt pathway is sensitive to salicylhydroxamic acid (SHAM), commonly used as an alternative oxidase inhibitor. Although SHAM affects not only alternative oxidase activity and can slightly modulate the Cyt pathway, it is conventionally accepted that SHAM-sensitive respiration is a measure for a contribution

of the Alt pathway to total respiration. The SHAM-resistant respiration corresponds to the sum of cytochrome-related electron transport and the residual non-mitochondrial respiration, which constitutes no more than 10% of total respiration [19].

Our study investigated the effect of short-term changes in temperature on respiration in intact tissues of onion leaves. We examined whether the $Q_{10}$ values of SHAM-resistant and SHAM-sensitive respiration differs and how shungite application to soil affects the temperature sensitivity of both SHAM-resistant and SHAM-sensitive respiratory pathways. Moreover, we established the extent to which the shungite dependence of SHAM-resistant and SHAM-sensitive pathways is affected by a change in the soil water availability.

## 2. Materials and Methods

### 2.1. Soil Substrate Preparation

The soil used in this study was collected from the 0–30cm topsoil layer of Umbric Podzols from the Korza valley in the northwest of Russia. Umbric Podzols, as have been stated earlier [20], are characterized by low natural fertility, a thin layer (10–20 cm), and low content of humus (0.5–2.5%). The soil used in this study has sandy loam with 19% of clay. As have been found recently [14], the soil has quite low water holding capacity and no evidence of influence of shungite application on the soil hydraulic characteristic was found. For this soil, the total C content was found to be $4.57 \pm 0.01$%, $pH_{KCl}$ was $5.46 \pm 0.03$, the N content was $0.39 \pm 0.02$%, the P and K content was $1.2 \pm 0.1$ and $0.15 \pm 0.01$ g kg$^{-1}$, respectively [14].

The soil was collected randomly, air-dried, and sieved with a 2 mm sieve. Shungite rock was taken from the Zazhogino deposit (Karelia, Russia) and crushed to a size of 0.5 mm. The entire volume of the dry soil was divided into four parts and mixed with shungite powder. Four concentrations of shungite powder were used in this experiment: 0, 5, 10, and 20 g of shungite per 1 kg of dry soil, designated as 0S, 5S, 10S, and 20S, respectively. These concentrations, close to natural conditions, were tested in a preliminary experiment and were used in the early study [14]. Before seed sowing, all soils were incubated under 21–23 °C and 70–80% of the maximum soil water holding capacity for 90 days.

### 2.2. Plant Growth Conditions

The soil substrates were parked into plastic pots (12 cm wide, 16 cm height). In this study we used onion seeds (*Allium cepa* L., var. Sturon) because this is a species of worldwide economic importance and, since onion roots are mainly accumulated in the upper soil layer, this species is sensitive to soil water limitation. Before sowing, uniform seeds of onion were imbibed in water for 3 h and sown with six seeds per pot. All pots were subjected to a controlled climate chamber (Vötsch BioLine, Balingen, Germany) with conditions of 23/20 °C day/night temperature, 70% relative air humidity, 16-h photoperiod, and 300 μmol m$^{-2}$ s$^{-1}$ of photosynthetic photon flux density. All pots were maintained and were well-watered for one week until seedlings were thinned to three seedlings per pot.

One week after sowing the pots of the 0S, 5S, and 10S treatments were randomly divided into two blocs and two watering treatments were applied: well watering (WW) and drying-wetting cycles (DW). The WW seedlings were watered daily to maintain the soil moisture content at the level of about 80% of the water holding capacity. The DW seedlings were watered once every five days: the soil was moistened to about 80% of the water holding capacity and then allowed to dry for 5 days. In this way, the completely randomized experimental design included four levels of shungite content in the soil (0S, 5S, 10S, and 20S) and two water regimes among the 0S, 5S, and 10S treatments. For the 20S treatment, only WW regime was applied. Each treatment included eight pots.

### 2.3. Total and SHAM-Resistant Respiration Measurement

Total respiration and SHAM-resistant respiration of leaves and roots were measured using a Clark-type oxygen electrode (Oxygraph Plus, Hansatech, Norfolk, UK). The mea-

surements were carried out at 20–22 days after the sowing using randomly selected, fully expanded green leaves. Before the onset of respiration measurements, the plants were kept in the dark for 15 min. A leaf sample (about 0.01 g of DW) was harvested with a razor blade, cut into small pieces, and suspended in 2 mL of air-saturated 100 mM Hepes buffer (pH 7.5) in the reaction vessel of the electrode unit. The $O_2$ uptake rate was measured in the presence of salicylhydroxamic (SHAM) acid, an agent commonly used as an inhibitor of alternative pathway respiration (Alt), or in the absence of the SHAM. The roots were carefully washed to remove soil and each whole root system was divided into two halves for the buffer with or without SHAM. The plant samples were kept in a buffer solution in darkness for approximately 15 min until the process rate was stabilized, and then the $O_2$ uptake rate was measured for 5 min. The rate of oxygen uptake by plant samples in a SHAM-free buffer solution was defined as total respiration ($V_t$); $O_2$ uptake rate in the SHAM-containing buffer was defined as SHAM-resistant respiration ($V_{SHAM-res}$), and the difference between $V_t$ and $V_{SHAM-res}$ was defined as SHAM-sensitive respiration ($V_{SHAM-sens}$). By neglecting the influence of SHAM on Cyt pathway activity, the contribution of $V_{SHAM-sens}$ to the $V_t$ rate (%) was calculated as $V_{SHAM-sens}/V_t$ ratio. While using this index, we realized the possible limitations of the adopted approach. We assumed that SHAM-sensitive $O_2$ uptake systems capable of mimicking the SHAM-inhibited respiration have a minor partitioning in the total oxygen uptake and are temperature-independent.

### 2.4. Temperature Response of $O_2$ Uptake Rates

To determine a temperature response of $V_t$, $V_{SHAM-res}$, and $V_{SHAM-res}$ respiratory pathways of onion leaves of $O_2$ uptake rates were measured at a buffer solution temperature of 13, 23, and 33 °C. The required temperature was attained by connecting the reaction vessel with a water-bath thermostat (VEB MLW Prufgerate-Werk, GDR). The Clark-type oxygen electrode calibration was carried out at each measurement temperature.

The temperature sensitivity of $O_2$ uptake rates was evaluated using the temperature coefficient ($Q_{10}$) that shows the proportional change in a respiration rate with a 10 °C increase in temperature. The $Q_{10}$ values were determined by approximating the plots of respiration rates at different temperatures with a power function.

### 2.5. Statistical Analysis

For each treatment, the means ± SE were determined with at least eight replicates. To assess the significant difference between the treatments, the least significant difference (LSD) of ANOVA was used at the $p < 0.05$ level. To ensure the normality and homogeneity of variances, the data were log-transformed if necessary. The effects of shungite concentration, water regime, and their interaction were analyzed using a two-way ANOVA for each measuring temperature separately. All statistical tests were carried out with Statistica software (v. 8.0.550.0, StatSoft, Inc., Tulsa, USA). When the differences between $O_2$ uptake rates in the absence and presence of SHAM were statistically insignificant, the $V_{SHAM-sens}$ value was assumed to be zero.

## 3. Results

### 3.1. Total and SHAM-Resistant Respiration

For the 0S seedlings, a significant impact of soil water deficit on total ($V_t$), but not SHAM-resistant ($V_{SHAM-res}$), respiration was found under low (13 °C) and high (33 °C) measurement temperatures (Figure 1a,c). The leaf $V_t$ rate was higher in 0S seedlings grown under DW, rather than under WW conditions. On the contrary, at 23 °C, no significant differences in the $V_t$ rates of leaves and roots were found between 0S seedlings grown under DW and WW regimes, but $V_{SHAM-res}$ was higher in DW than WW leaves (Figure 1b). According to the two-way ANOVA, the $V_{SHAM-res}$ rate of both leaves and roots was significantly affected by soil water availability, but the effect of shungite application was not significant for both $V_t$ and $V_{SHAM-res}$ rates at all measurement temperatures (Table 1). However, for certain conditions of temperature and soil water availability, this effect was

significant enough. Therefore, the shungite application decreased leaf $V_t$ and $V_{SHAM-res}$ under DW and increased $V_t$ under WW regime at 13 °C (Figure 1a). Moreover, under the WW regime and at the temperatures of 23 and 33 °C, seedlings grown on the soil containing shungite had lower $V_{SHAM-res}$ values than 0S seedlings (Figure 1b, c). For the roots, both $V_t$ and $V_{SHAM-res}$ rates of 5S and 10S seedlings were lower under DW and higher under WW than 0S seedlings, but these differences were not large enough to be statistically significant (Figure 1d).

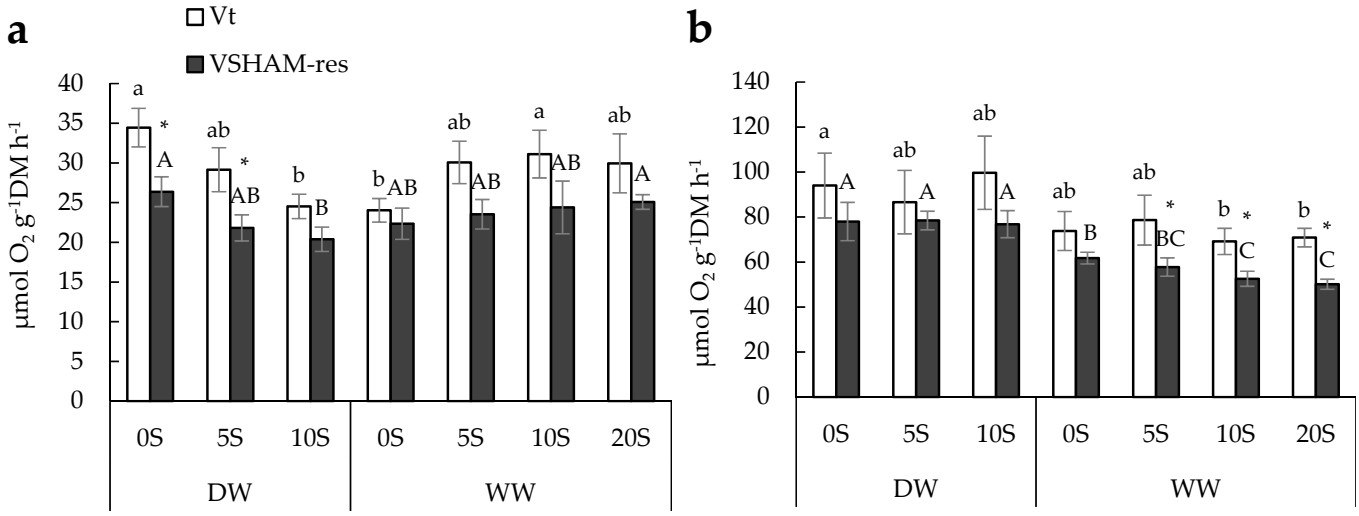

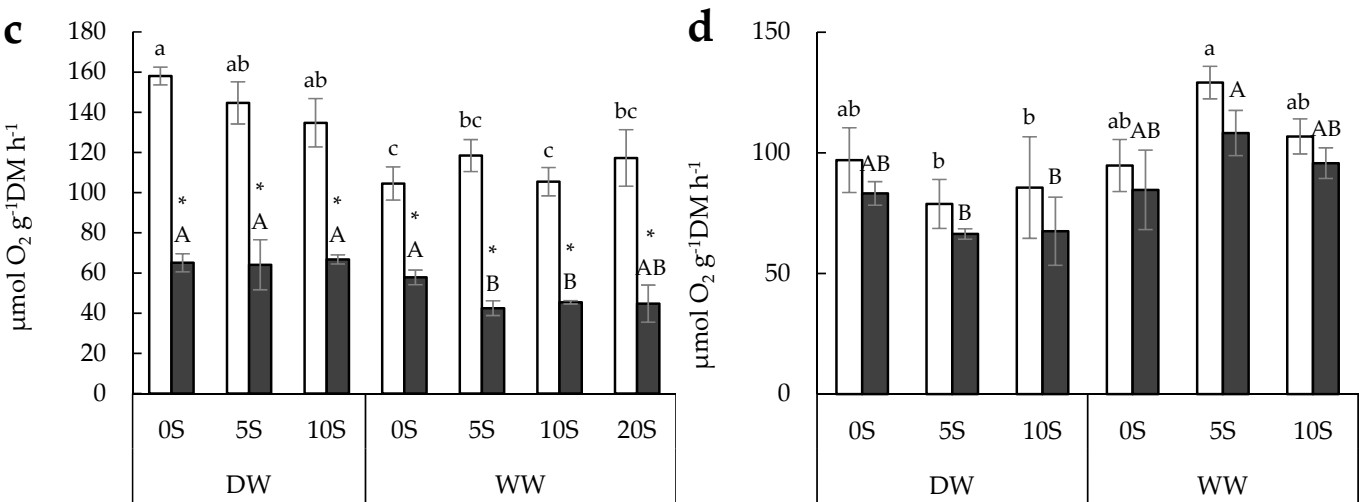

**Figure 1.** Total respiration ($V_t$) and SHAM-resistant respiration ($V_{SHAM-res}$) for onion leaves (**a–c**) and roots (**d**) grown on the Umbric Podzols with shungite concentration of 0 (0S), 5 (5S), 10 (10S), and 20 (20S) g kg$^{-1}$ under drying-wetting (DW) or well watering (WW) regime. During the measurements, the temperature was kept at 13 (**a**), 23 (**b**,**d**), or 33 (**c**) °C. Different letters indicate significant differences between mean values under different soil shungite content and water regimes. * indicates significant differences between $V_t$ and $V_{SHAM-res}$ at $p < 0.05$.

**Table 1.** Statistical results (*p*-value) of two-way ANOVA for the parameters shown in Figures 1 and 2.

| Variables | Treatment Factor, Interaction | | |
| --- | --- | --- | --- |
| | Shungite | Water Regime | Shungite + Water Regime |
| Leaves | | | |
| 13 °C | | | |
| $V_t$ | 0.233 ns | 0.194 ns | 0.052 ns |
| $V_{SHAM-res}$ | 0.310 ns | 0.192 ns | 0.118 ns |
| $V_{SHAM-sen}/V_t$ | <0.001 *** | 0.702 ns | 0.047 * |
| 23 °C | | | |
| $V_t$ | 0.606 ns | 0.093 ns | 0.420 ns |
| $V_{SHAM-res}$ | 0.231 ns | <0.001 *** | 0.582 ns |
| $V_{SHAM-sen}/V_t$ | 0.049 * | 0.039 * | 0.030 * |
| 33 °C | | | |
| $V_t$ | 0.551 ns | 0.093 ns | 0.394 ns |
| $V_{SHAM-res}$ | 0.231 ns | <0.001 *** | 0.440 ns |
| $V_{SHAM-sen}/V_t$ | <0.001 *** | <0.001 *** | 0.621 ns |
| Roots | | | |
| 23 °C | | | |
| $V_t$ | 0.788 ns | 0.042 * | 0.161 ns |
| $V_{SHAM-res}$ | 0.902 ns | 0.022 * | 0.216 ns |
| $V_{SHAM-sen}/V_t$ | 0.405 ns | 0.108 ns | 0.621 ns |

$V_t$, total respiration; $V_{SHAM-res}$, SHAM-resistant respiratory pathway; $V_{SHAM-sen}$, SHAM-sensitive respiratory pathway. Asterisks denote significance levels: * $p < 0.05$, *** $p < 0.001$; ns, not significant.

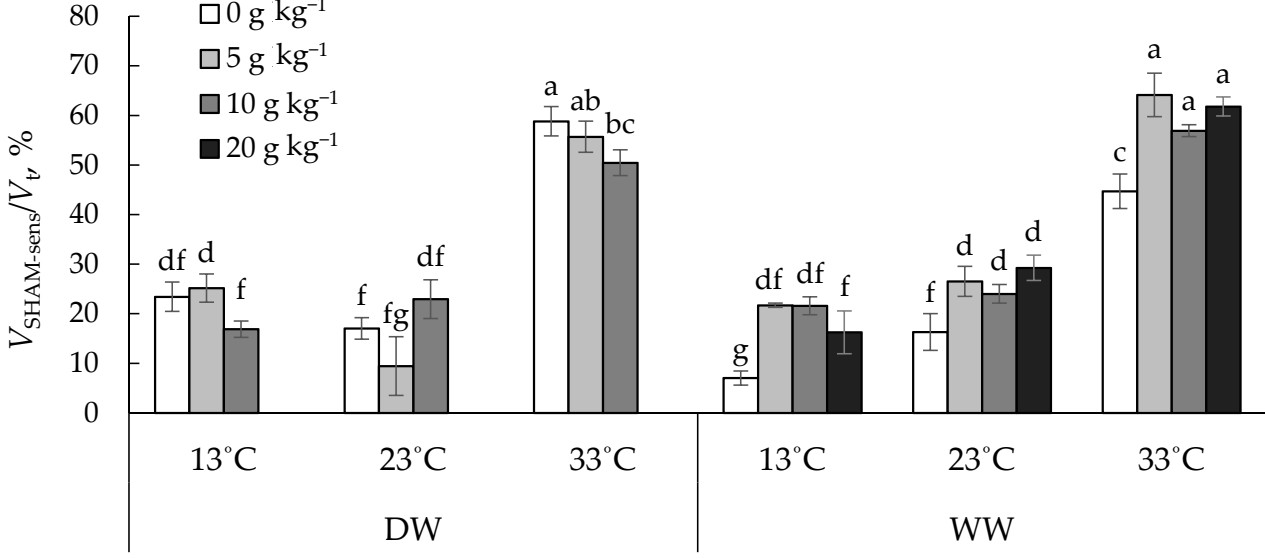

**Figure 2.** The $V_{SHAM-sen}/V_t$ ratio for leaf respiration of onion seedlings, grown on the Umbric Podzols with shungite concentration of 0, 5, 10, and 20 g kg$^{-1}$ under drying-wetting (DW) or well watering (WW) regime. Different letters indicate significant differences between mean values under different soil shungite contents and water regime.

### 3.2. Ratio of SHAM-Sensitive to Total Respiration

Onion leaf respiration was more sensitive to the inhibitor of the alternative path (salicylhydroxamic acid, SHAM) than root respiration, resulting in the decrease of leaf respiration when $O_2$ uptake was measured in the presence of SHAM (Figure 1). However, this decrease was not significant for all combinations of the measurement temperature, soil water regime, and shungite treatment. Therefore, at 13 °C, leaf respiration was more sensitive to the SHAM for DW than WW seedlings, but at 23 °C, on the contrary, it was more sensitive for WW than DW seedlings.

The increase of respiration sensitivity to the SHAM reflects an increase in the contribution of Alt respiratory pathway to total respiration. With the increase of the measurement temperature, the $V_{SHAM-sens}/V_t$ ratio tended to increase (Figure 2). For the 0S leaves,

$V_{SHAM-sens}/V_t$ values were higher in DW than WW seedlings regardless of the measurement temperature.

In contrast to the roots, for the leaves, the two-way ANOVA revealed a significant effect of both shungite application and soil water availability and their interaction on the $V_{SHAM-sen}/V_t$ ratio (Table 1). In contrast to DW cycle, shungite application significantly increased the leaf $V_{SHAM-sen}/V_t$ values, for the seedlings grown under WW regime at all measurement temperatures regardless of soil shungite content (Figure 2).

### 3.3. Respiratory Coefficient ($Q_{10}$)

The $Q_{10}$ values of leaf $V_t$ and $V_{SHAM-res}$ did not differ significantly between the 0S seedlings grown under DW and WW regimes, but $Q_{10}$ of $V_{SHAM-sens}$ decreased strongly following the decrease in soil water availability (Figure 3). Regardless of shungite treatment and soil water regime, the $Q_{10}$ values were higher for the $V_{SHAM-sens}$, than $V_t$ and $V_{SHAM-res}$ rates. The opposite effect of shungite application on respiratory coefficient was found for seedlings grown under DW and WW regimes. While under DW conditions, shungite application increased $Q_{10}$ of both $V_{SHAM-res}$ and $V_{SHAM-sens}$, increasing $V_t$, under the WW regime, shungite significantly decreased respiratory coefficient of both respiratory pathways.

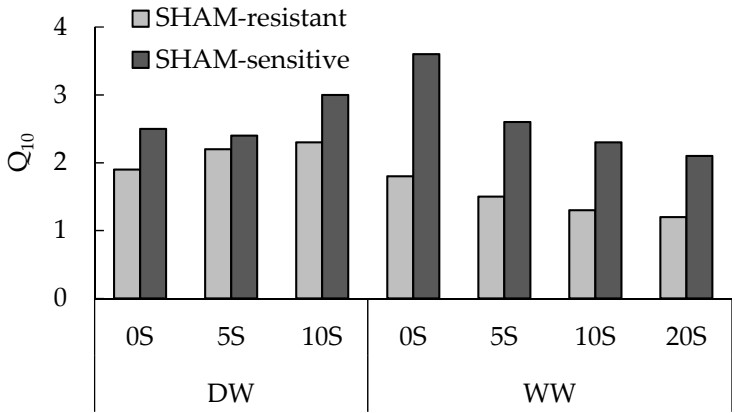

**Figure 3.** Temperature sensitivity ($Q_{10}$) of the SHAM-resistant and SHAM-sensitive leaf respiration of onion seedlings growing on the Umbric Podzols with shungite concentration of 0 (0S), 5 (5S), 10 (10S), and 20 (20S) g kg$^{-1}$ under drying-wetting (DW) or well-watered (WW) regime.

### 4. Discussion

This study was aimed at assessing whether shungite application to soil have the potential to alter the leaf and root respiration rate and temperature sensitivity ($Q_{10}$) of SHAM-resistant respiration, reflecting mainly the Cyt pathway, and SHAM-sensitive respiration, reflecting the Alt pathway, using onion seedlings as a model. Therefore, we quantified the effects of soil water availability on the respiratory $Q_{10}$ to understand whether its response to shungite application depends on the soil water conditions. The onion leaves and roots showed a different response to the soil water deficit (Figure 1). While among all treatments the total root respiration tended to decrease, leaf respiration increased during water limitation. Moreover, leaves were more sensitive to both shungite application and SHAM: onion roots were insensitive to the effects of neither shungite nor acid.

The positive role of the Alt pathway in plant metabolism under stress conditions has been widely discussed [10–13] due to its ability to stabilize the reduction level of the ubiquinone pool and prevent the production of excessive amounts of reactive oxygen species [21]. Soil water deficit and low temperature are some of the important factors limiting crop yield. We have shown that soil water deficit can stimulate onion leaf respiration mainly due to an increased rate of SHAM-sensitive respiration (Figure 1). While our results (Figure 2) support previous findings that the contribution of Alt pathway to total respiration increases following decreased water availability [22], for onion leaves, this was

only found at the low and high measurement temperature. However, at the optimal temperature, the increase of total leaf respiration in response to water limitation was associated with the increase of SHAM-resistant respiration. Thus, different mitochondrial respiratory pathways may be involved to plant respiration acclimation to soil water deficit with the pathway contribution depending on environmental conditions, particularly temperature.

Our finding (Figure 3) that the temperature sensitivity of SHAM-sensitive respiration is higher than the sensitivity to short-term changes in temperature for SHAM-resistant respiration is consistent with earlier studies [17,18]. It is known that plant respiratory $Q_{10}$ values are affected by the growth environment and may vary significantly [17]. While the temperature sensitivity of respiration has been mainly studied at the total respiration level, the results of this study highlight the variability of both SHAM-resistant and SHAM-sensitive respiratory pathways. The results have shown that the $Q_{10}$ values of studied respiratory pathways of onion leaves can be altered by changes in soil water availability, as well as shungite application to the soil. Slot et al. [23] found that the $Q_{10}$ of total dark respiration of *Geum urbanum* leaves, as well as the respiration rate, increases in response to the decrease in water availability. For onion leaves, our results have shown the same effect of water deficit on total respiration, although in our study the sensitivity of total respiration to short-term temperature changes was not strongly affected by the soil water regime (data not shown). However, water deficit slightly increased the $Q_{10}$ value of SHAM-resistant respiration and significantly decreased this parameter of the SHAM-sensitive pathway, as was found for seedlings grown on the soil without shungite (Figure 3, 0S treatment).

The effect of shungite application on the temperature sensitivity of SHAM-resistant and SHAM-sensitive respiration of onion leaves was strongly dependent on the soil water availability. While the shungite application increased the $Q_{10}$s of both SHAM-resistant and SHAM-sensitive respiration during water deficit, shungite decreased these values under the condition of sufficient water availability. The cause of the variability in respiratory $Q_{10}$ values has not been well established yet, especially for the respiratory pathways. The temperature sensitivity of respiratory flux has been shown to be variable, depending on the level of ubiquinone reduction, the degree of adenylate control of the Cyt pathway [17], and/or availability of respiratory substrates [24]. The $Q_{10}$ of $O_2$ consumption values can increase following the substrate availability increase [24], an increase of ubiquinone reduction, and an increase of activation state of the Alt pathway [17]. A recent study [14] showed that the shungite application to the soil can alter nutrient concentrations of onion seedlings. Depending on the soil water regime, shungite can increase the plant content of potassium, manganese, zinc, and nickel, thus affecting some physiological traits of onion seedlings [14]. It can be suggested that the shungite-related change of $Q_{10}$s of SHAM-resistant and SHAM-sensitive respiration might be partly controlled by the nutrient element contents.

The ability of plants to enhance electron transport through the Alt pathway in the cold [10,11,13,25] could be due to the Alt pathway being less temperature-sensitive than the Cyt pathway [26]. The results of this study are consistent with some earlier reports, which have shown that the sensitivity of Alt to the short-term temperature changes may not be lower than that of Cyt [17,18,25]. For the 0S, onion leaves grown under the condition of sufficient water availability, the higher $Q_{10}$ of the Alt pathway than the Cyt pathway was connected with the lowest contribution of SHAM-sensitive respiration to total respiration at a low measurement temperature (Figure 2). These data confirm the findings of Armstrong et al. [18] that during the short-term temperature drop the activity of electron transport through the Alt pathway declines and does not play an important role in maintaining flux through the mitochondrial electron transport. However, both shungite application at sufficient water availability and soil water deficit may decrease the sensitivity of the SHAM-sensitive pathway to the short-term temperature change. This allows the Alt pathway to be more involved in the electron transport process and in doing so reduces the production of reaction $O_2$ species. Under the condition of the well-watered regime, the shungite application decreased the temperature sensitivity of not only the SHAM-sensitive, but

also the SHAM-resistant pathway (Figure 3). The decrease of temperature sensitivity of respiratory pathways can help to maintain mitochondrial electron transport and cell redox-state during the temperature drop, and thus the plant resistance, to a low temperature.

## 5. Conclusions

The data demonstrate that both SHAM-resistant and SHAM-sensitive respiratory pathways of *A. cepa* leaves and their sensitivity to short-term temperature change can be dynamic when plants are subjected to the contrasting conditions of soil water availability or shungite content. For plants grown without shungite, the water deficit decreased the $Q_{10}$ values of SHAM-sensitive, but not SHAM-resistant, respiration. The response of the temperature sensitivity of the pathways to shungite application depends on the water availability. The shungite-related decrease of both SHAM-resistant and SHAM-sensitive pathways may play an important role in enhancing the resistance of plant respiration to the temperature drop.

Although shungite rocks are widely used in various industries, their use in agriculture is still being studied. The obtained results showed that the shungite rocks might have the potential for agricultural application, although further investigations, including field studies, are required.

**Author Contributions:** E.I. designed and performed the experiments, analyzed the data, wrote the paper; S.C. wrote the paper; O.B., V.S. conceived the experiments. All authors have read and agreed to the published version of the manuscript.

**Funding:** This research was funded by the project of the Ministry of Science and Higher Education of the Russian Federation, grant number 0218-2019-0074 and AAAA-A18-118020690231-1, and partly by the Russian Foundation for Basic Research, grant number 19-29-05174.

**Institutional Review Board Statement:** Not applicable.

**Informed Consent Statement:** Not applicable.

**Data Availability Statement:** Not applicable.

**Acknowledgments:** Experimental facilities for this study were offered by the Core Facility of the Karelian Research Centre of the Russian Academy of Sciences.

**Conflicts of Interest:** The authors declare no conflict of interest. The founding sponsors had no role in the design of the study; in the collection, analyses, or interpretation of data; in the writing of the manuscript, and in the decision to publish the results.

## Abbreviations

The following abbreviations are used in this manuscript:

| | |
|---|---|
| Alt | alternative pathway |
| Cyt | cytochrome c oxidase (Cyt) pathway |
| DW | drying-wetting cycle |
| RAS | Russian Academy of Science |
| SHAM | salicylhydroxamic acid |
| WW | well-watered |

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
