# Peer review of "Effect of Shungite Application on the Temperature Sensitivity of Allium cepa Respiration under Two Soil Water Regimesâ€"

_agronomy, doi:10.3390/agronomy11071302_

Round 1
Reviewer 1 Report
Dear Authors
The manuscript discusses whether the shungite application to soil may affect leaf and root mitochondrial respiratory pathways, and leaf response to a temperature which could be an interesting topic to the readers of the Journal. few comments are given below.
1) what was the basis for the determination of the concentration of shungite ? or are they just arbitrary values? because the lowest concentration may require about 15 tons/ha in real field applications. would that be practical? what is the availability of the rock powder? I suggest the authors add some information on this to the discussion.
2) what are the characteristics of the soil, for example, soil texture, organic matter content, water holding capacity, if the data available, I suggest authors add a table of them. this will be important to understand the soil response to the rock powder.
3) It is not that clear how the watering treatments were imposed. for example, was the water added up to again 80 % of WHC of soil and let dry for 5 days? I suggest the authors add more details on this.
4) I suggest improving the conclusion section by adding the real applications of the results. That would be of much interest to the readers.
Author Response
First we would like thank the Reviewers for attentive revision of our manuscript and constructive comments and suggestions.
Let me indicate the modifications made in the manuscript in the light of Reviewer’s comments.
- what was the basis for the determination of the concentration of shungite ? or are they just arbitrary values? because the lowest concentration may require about 15 tons/ha in real field applications. would that be practical? what is the availability of the rock powder? I suggest the authors add some information on this to the discussion.
Thank You. Shungite concentrations were tested in a preliminary experiment and used in our early experiment (Ikkonen et al., 2021, Acta Physiologia Plantarum). This shungite content under study is close to natural field conditions for shungite soils, found in Karelia area.
So, we added to the text “These concentrations, close to natural conditions, were tested in a preliminary experiment and used in the early study [14]. ( in 2. Materials and Methods, 2.1. Soil substrate preparation)
Shungite rocks are widely distributed in Karelia and they preserved giant amount of elemental carbon estimated as more than 25·1010 tones (Buseck et al., 1997). The shungite deposits have been intensively mined since 1960s, because the shungite rocks are used in various industries such as metallurgy, construction, chemical industry.
So, we added to the text:
“The amount of elemental carbon preserved in shungite rocks was estimated to be more than 25·1010 tones [5].” (yellow in the Introduction)
- what are the characteristics of the soil, for example, soil texture, organic matter content, water holding capacity, if the data available, I suggest authors add a table of them. this will be important to understand the soil response to the rock powder.
Thank You for this important comment. We added to the text:
“The soil used in this study has sandy loam with 19% of clay. As have been found recently [14], the soil has quite low water holding capacity and no evidence of influence of shungite application on the soil hydraulic characteristic was found. For this soil, the total C content was found to be 4.57 ± 0.01%, pHKCl was 5.46 ± 0.03, the N content was 0.39 ± 0.02%, the P and K content was 1.2 ± 0.1 and 0.15 ± 0.01 g kg-1, respectively [14].” (yellow in Materials and Methods, 2.1. Soil substrate preparation).
- It is not that clear how the watering treatments were imposed. for example, was the water added up to again 80 % of WHC of soil and let dry for 5 days? I suggest the authors add more details on this.
Thank You, You are right. We changed the test to:
“The DW seedlings were watered once every five days: the soil was moistened to about 80% of the water holding capacity and then allowed to dry for 5 days” (yellow in Materials and Methods, 2.1. Soil substrate preparation).
- I suggest improving the conclusion section by adding the real applications of the results. That would be of much interest to the readers.
We added to the conclusion:
“Although shungite rocks are widely used in various industries, their use in agriculture is still being studied. The obtained results showed that the shungite rocks might have the potential for agricultural application though the further investigations including the field studies are required.” (yellow)
The changed text in manuscript is now colored with yellow.
Thanks one more!
Best regards,

Reviewer 2 Report
The manuscript is well prepared and meets the requirements outlined in the Agronomy. Overall, I think that the presented work is attractive and the manuscript has enough scientific content. Anyhow, I would suggest it be published after minor revisions as below:
- Line19 I suggest adding information about two regimes of temperature because it is one of the experimental conditions.
- Line 103 What was the reason for a chosen Allium Cepa to the research?
- Line 139 Why the treatment of 20S was applied only for the WW regime?
- All figures: Please add the legends. I think that it will be more clear.
- Figure 1. I mean that different letters indicate significant differences between levels of shungite in the soil. Please clarify this in the label
- Figure 2. There is a lack of y-axis description for figure 2(b)
Author Response
First we would like thank the Reviewer for attentive revision of our manuscript and constructive comments and suggestions.
Let me indicate the modifications made in the manuscript in the light of Reviewer’s comments.
- Line19 I suggest adding information about two regimes of temperature because it is one of the experimental conditions.
Thank You. Only one temperature regime for the seedlings growth was use in this study - 23/20°C day/night temperature – as is mentioned in 2.2. Plant growth conditions.
Three levels of the temperature (13, 23, and 33°C) were used only during measurements to determine a response of Vt and VSHAM-res respiratory pathways to short-term tempetature change. (2.3. Temperature response of O2 uptake rates).
Line 103 What was the reason for a chosen Allium Cepa to the research?
Allium cepa seedlings were chosen this study because most of the onion roots are accumulated in the upper soil layer, so this species is sensitive to soil water deficit and very informative in the studies with soil water deficit conditions. Moreover, onion plants are actively cultivated species in Northern Europe.
So, we added to the text:
“In this study we used onion seeds (Allium cepa L., var. Sturon) because this is species of worldwide economic importance and, since onion roots are mainly accumulated in the upper soil layer, this species is sensitive to soil water limitation”.
- Line 139 Why the treatment of 20S was applied only for the WW regime?
Thank You. For the treatment 20S, we were forced to use only WW regime due to the limited area of the climatic chamber.
- All figures: Please add the legends. I think that it will be more clear.
Thank You. Done.
- Figure 1. I mean that different letters indicate significant differences between levels of shungite in the soil. Please clarify this in the label
Thank You. Done.
- Figure 2. There is a lack of y-axis description for figure 2(b)
Thank You. Done.
The changed text in manuscript is now colored with yellow.
Thanks one more to you!
Best regards,
